# Scattered Tubular Cells Markers in Macula Densa of Normal Human Adult Kidney

**DOI:** 10.3390/ijms231810504

**Published:** 2022-09-10

**Authors:** Giovanni Tossetta, Sonia Fantone, Teresa Lorenzi, Andrea Benedetto Galosi, Andrea Sagrati, Mara Fabri, Daniela Marzioni, Manrico Morroni

**Affiliations:** 1Department of Experimental and Clinical Medicine, Università Politecnica delle Marche, 60126 Ancona, Italy; 2Department of Clinical and Specialist Sciences, Clinic of Urology, Università Politecnica delle Marche, 60126 Ancona, Italy; 3Department of Life and Environmental Sciences, Università Politecnica delle Marche, 60131 Ancona, Italy; 4Electron Microscopy Unit, Azienda Ospedaliero-Universitaria Ospedali Riuniti, 60126 Ancona, Italy

**Keywords:** scattered tubular cells, macula densa, normal human kidney, prominin-1 (CD133), cytokeratin 7, vimentin, immunohistochemistry

## Abstract

Background: The scattered tubular cells (STCs) are a population of resident progenitor tubular cells with expansion, self-renewal and epithelial differentiation abilities. Although these cells are localized within the proximal (PTs) and distal (DTs) tubules in a normal adult kidney, their presence has never been demonstrated in human macula densa (MD). The purpose of the present study is to describe the presence of STCs in MD using specific markers such as prominin-1 (CD133), cytokeratin 7 (KRT7) and vimentin (VIM). Methods: We analyzed two sets of three consecutive serial sections for each sample. The first sections of each set were immunostained for nNOS to identify MD, the second sections were immune-stained for CD133 (specific STCs marker) while the third sections were analyzed for KRT7 (another STCs specific marker) and VIM (that stains the basal pole of the STCs) in the first and second sets, respectively, in order to study the co-expression of KRT7 and VIM with the CD133 marker. Results: CD133 was localized in some MD cells and in the adjacent DT cells. Moreover, CD133 was detected in the parietal epithelial cells of Bowman’s capsule and in some proximal tubules (PT). KRT7-positive cells were identified in MD and adjacent DT cells, while KRT7 positivity was mostly confined in both DT and collecting ducts (CD) in the other areas of the renal parenchyma. CD133 and KRT7 were co-expressed in some MD and adjacent DT cells. Some of the latter cells were positive both for CD133 and VIM. CD133 was always localized in the apical part of the cells, whereas the VIM expression was evident only in the cellular basal pole. Although some cells of MD expressed VIM or CD133, none of them co-expressed VIM and CD133. Conclusions: The presence of STCs was demonstrated in human adult MD, suggesting that this structure has expansion, self-renewal and epithelial differentiation abilities, similar to all other parts of renal tubules.

## 1. Introduction

The macula densa (MD) cells are renal sensory elements belonging to the juxtaglomerular apparatus (JGA) that plays an important role in regulating renal blood flow, glomerular filtration rate and renin release. The MD cells have a primary cilium in their apical membrane exposed to the tubular fluid and a basolateral portion in close proximity to the effector cells of the JGA (i.e., the renin-producing granular cells of both afferent and efferent arterioles) and to the contractile cells of the extraglomerular mesangium. The MD position is strategic, allowing it to detect the alteration of tubular fluid components (e.g., the ionic composition) and to act on the JGA effectors by synthesizing and releasing paracrine mediators. It is known that increased NaCl tubular concentration induces adenosine triphosphate release at the basolateral portion of MD cells, causing afferent arteriole vasoconstriction and consequently producing glomerular filtration rate (GFR) reduction (tubuloglomerular feedback), directly or through breakdown to adenosine [1,2,3,4,5,6]. This negative feedback prevents GFR fluctuation and helps to regulate both the renal blood flow and the GFR. In addition, this mechanism is specifically directed to stabilize NaCl absorption from the distal tubule (DT). Most of the morphological observations of the MD have been performed in animal models [7,8,9,10,11], while few studies have been performed on humans [8,12], including our recent morphological studies [13,14]. Previous observations show that the MD comprise a few cells, about 15–20 cells per nephron [15,16]. Contrarily, we observed that the MD is a monolayer cap of 35–40 cells arranged in five to six layers and extended for about 40 µm using toluidine blue-stained serial semithin sections. MD cells showed two specialized ends at the ultrastructural level: the apical end, which is equipped with a primary apical cilium, and the basolateral end showing morphological features of a paracrine secretory cell (for the presence of numerous small dense granules). Apical tight junctions connected MD cell membranes, forming a tubule-mesangial barrier. The scattered tubular cells (STCs) are a population of progenitor cells localized in both the proximal (PT) and distal (DT) tubule in normal adult kidneys. STCs can be considered as resident progenitor cells having expansion, self-renewal and epithelial differentiation abilities. Interestingly, the simultaneous presence of both degenerating and undifferentiated cells similar to STCs in MD and in the adjacent DT cells, as previously identified by electron microscopy, can suggest that the MD is a dynamic tissue undergoing self-renewal [14].

The aim of this study was to confirm the presence of STCs in both MD (Figure 1; in orange) and adjacent DT cells (Figure 1; in yellow) of the adult normal human kidney, using a set of specific immunohistochemical markers such as prominin-1 (CD133), cytokeratin 7 (KRT7) and vimentin (VIM). STC markers identification in MD by our morphological analysis provides evidence that both cell degeneration and cell differentiation processes are present in this small area of renal parenchymal, suggesting MD’s self-renewal capacity.

## 2. Results

### 2.1. CD133 and KRT7 Expression

CD133 was expressed in some MD cells and in the adjacent DT cells. An evident CD133 staining was present in the cellular apical part as a bow surrounding the nucleus (Figure 2b). CD133 was also expressed in the parietal epithelium of Bowman’s capsule and in some proximal tubule (PT) and DT cells (positive internal control) (Figure 3a). CD133-positive cells were frequently localized in tubular creases, and sometimes appeared in duplets or in cluster of three to four positive cells (Figure 3b). Renal glomeruli were negative (negative internal control) (Figure 2b). KRT7-positive cells were identified in MD and adjacent DT cells, while they were mostly confined in both DT and collecting ducts (CD) in other areas of the renal parenchyma. A single PT cell or very few PT cells were KRT7-positive. Renal glomeruli were mainly negative, but short segments of the capsular epithelium showed KRT7-positivity in some corpuscles (Figure 4).

Interestingly, CD133 and KRT7 were co-expressed in some MD and adjacent DT cells (Figure 2b,c).

### 2.2. Some DT Cells Adjacent to MD Co-Express CD133 and VIM

The DT adjacent to MD showed some cells positive for both CD133 and VIM. CD133 was always localized in the cellular apical part, whereas VIM was evidently expressed only in the cellular basal pole (under the nucleus) (Figure 5b,c). VIM and CD133 co-expression was not observed in any MD cell, while cells positive for VIM or for CD133 were found in the MDs (Figure 5b,c). Renal glomeruli (including the capsular epithelium), vessels, interstitium, and basal cytoplasm of some PT and DT (positive internal control) were positive for VIM (Figure 6b). Cells positive for CD133 but negative for VIM, and vice versa, were noted in PT and DT (Figure 6a,b).

## 3. Discussion

We demonstrated, for the first time, that MD cells and the DT cells adjacent to MD express such STCs markers as CD133, KRT7 and VIM in adult normal human kidneys [17,18,19,20]. The STCs are a population of progenitor cells localized within the PTs and DTs in normal adult kidneys, which have a more pronounced regenerative potential than differentiated tubular epithelial cells [21,22,23]. STCs were initially detected in rodent kidneys [23] and then in humans [17,24,25], while STC markers were also expressed in the parietal epithelium of Bowman’s capsule [17,24,26]. It was demonstrated that STCs can be considered resident progenitor cells showing expansion, self-renewal and epithelial differentiation abilities using both in vitro and in vivo models [24,25,27]. Their morphology is in accordance with the function of progenitor cells, including basally located filaments closely associated to the cytoplasmic membrane. These cells lack both microvilli and basolateral infoldings (Figure 7) [14,24]. Some observations evidence that STCs show a higher proliferation index, becoming more numerous after acute kidney injury [25]. However, it is not still clear if these cells represent a steady progenitor population or a population with transient regenerative phenotype [28,29], and if MD cells undergo turnover. The lack of Bromodeoxyuridine (BrdU) and Ki67 immunostaining in MD cells excluded the proliferative ability of these cells [30,31]. Interestingly, some studies performed on rats treated with candesartan, an angiotensin II type-1 receptor antagonist, suggested that the increased number of MD cells may be caused by normal tubular cells’ trans-differentiation into MD cells [30].

We previously found both degenerative and immature cells in MD and in DT cells adjacent to MD in adult normal human kidneys by electron microscopy [14]. In that study, we identified the immature cells as STC elements due to their ultrastructural features and, as demonstrated in this study, to their CD133, KRT7 and VIM expression. Thus, the present data, obtained by analyzing 31 MDs, confirm the findings of our previous study on 3 MDs [14]. In addition, we and others have shown that these cells did not express the proliferative marker Ki67 [14,30] and apoptotic markers such as caspase-3 and caspase-9 [14]. We hypothesized that the MD cells and the adjacent DT cells also undergo substantial renewal due to the important functional task to which they are subjected, the same as for the PT cells. The latter are more susceptible to injury, but at the same time have a remarkable ability to restore their morphology and function [32,33].

STCs are detected in normal human MD from the postnatal period (1–5 years of age) [34] to adulthood (present study), the same as for the PT [24], suggesting that any damage to the MD during its functional activity could be repaired by cell progenitor recruitment normally present in the tissue at any age. Recent findings suggest that keratins may be required to maintain the balance between proliferation and differentiation processes [35]; therefore, we cannot exclude that the high KRT7 expression in these cells is due to the differentiation process of STCs in this region of the renal parenchyma. CD133 and KRT7 co-expressing cells can be considered progenitor cells that could start cell differentiation towards both MD and adjacent DT cells. Moreover, stress-induced keratin upregulation occurs in many tissues at the transcriptional level and during the regenerative phase after injury, suggesting that keratins (including KRT7) are important in tissue repair [36] and can be considered markers and regulators of renal tubular epithelial injury [37]. 

We previously demonstrated that some filaments were present in the basal pole of STCs, both in MD and in adjacent DT cells, by ultrastructural analysis [14], and others showed that these filaments were positive for VIM [24]. So, we investigated whether STCs positive for CD133 were also positive for VIM in MD cells. Our data confirm that some STCs positive for CD133 also expressed VIM, but only in the DT cells adjacent to the MD, while no co-staining was present in MD cells, although cells positive only for VIM were observed. This discrepancy of co-expression between CD133 and VIM can be related to the tangential cutting plane of the STCs, demonstrated by the presence of CD133-positive and VIM-negative cells in both PT and DT, and vice versa. It has been suggested that basal VIM is related to increased mechanical resilience in the STCs, which makes them more robust [24], and that it is also regarded as an anti-apoptotic factor [38], being expressed in regenerating renal tubules following ischemic injury in both humans and rats [39].

To date, it is not known how MD changes under conditions of kidney disease, such as in the course of glomerulonephritis or tubular damage, also because, as previously described, this tissue is difficult to identify, since it consists of few cells. In this study, we showed that MD contains a stem cell niche, increasing the current knowledge of its biology, and opening up new possible therapeutic scenarios aimed at studying the control of the renal blood flow, the glomerular filtration rate, and the renin release modulating the arterial pressure and the tubule–glomerular feedback.

## 4. Materials and Methods

### 4.1. Paraffin Samples

The procedures followed for the collection of samples were in accordance with the Helsinki Declaration of 1975, as revised in 2013. All patients provided their written informed consent, all information regarding human material was managed using anonymous numerical codes, and all samples were handled in compliance with the Helsinki Declaration as revised in 2013. All normal-looking tissue samples were selected from renal samples routinely processed for diagnostic purpose. All patients had a normal renal function and no history of kidney disease at the time of sampling (Table 1).

The MD is about 40 µm long, and each cell that composes the MD is 8–12 µm high and 7–8 µm wide [14]. Therefore, the same cell belonging to the MD could be observed in two different parallel sections, each with a thickness of 4 µm when the best conditions exist. So, we performed two consecutive sets of sections, containing three parallel sections each for each sample, in order to identify the same cell at least in two consecutive sections. The first section of each set aimed to identify MD by studying nNOS, which is the immunohistochemical specific marker [40], whereas the other two sections of each set allowed us to study the expression of two different STCs markers. In particular, in the first set, CD133 and KRT7, while in the second set CD133 and vimentin, expressions were examined. VIM was studied since the basal filaments of the STCs are positive for this molecule [24], and this permits us to identify the same MD cell positive for CD133. Serial sections of 4 μm in thickness (total thickness about 12 μm) were cut as above described and stretched at 45 °C, then placed to dry and stored at 4 °C until use. A schematic representation of the serial section setting is shown in Figure 8.

### 4.2. Immunohistochemical Analysis

Immunohistochemical staining was performed as previously described [41,42]. Briefly, sections were deparaffinized and rehydrated through xylene and a graded series of ethyl alcohol. To inhibit endogenous peroxidase activity, sections were incubated with 3% hydrogen peroxide (in deionized water) for 1 h at RT, then washed in phosphate-buffered saline (PBS), immersed in 0.1 M citrate buffer pH 6 and subjected to high-temperature treatment for 10 min at 98 °C. After washed in PBS, all sections were incubated for 1 h at RT with goat normal serum (Vector laboratories, Burlingame, CA, USA) diluted 1:75 in PBS to block nonspecific background, and then overnight at 4 °C with the primary antibodies listed in Table 2 diluted in PBS. After washing in PBS, sections were incubated for 1 h at RT with the appropriate secondary biotinylated antibody (Vector Laboratories) diluted 1:200 in PBS. The peroxidase avidin–biotin complex method (Vector Laboratories) was applied for 1 h at RT using 3′,3′ diaminobenzidine hydrochloride (DAB; Sigma, Milano, Italy) as chromogen. Sections were counterstained with Mayer’s hematoxylin, dehydrated, and finally mounted with Eukitt solution (Kindler GmbH and Co., Freiburg, Germany). Negative controls were performed by omitting the primary or the secondary antibody. Specific isotypic control antibodies were used as a further negative control at the same dilutions and under the same conditions as the primary antibodies (Table 2).

### 4.3. Transmission Electron Microscopy

The ultrastructural features of the STCs were assessed by electron microscopy. Renal specimens were fixed in 2% glutaraldehyde/2% paraformaldehyde in 0.1 M phosphate buffer for 3 h at 4 °C, postfixed in 1% osmium tetroxide in the same buffer solution, dehydrated in graded alcohols, and embedded in an Epon-Araldite mixture. Semithin sections (2 μm) were obtained from each specimen with a MICROM HM 355 microtome (Zeiss, Oberkochen, Germany) and stained with toluidine blue. Thin sections were cut using an MTX ultramicrotome (RMC, Tucson, AZ, USA), stained with lead citrate, and examined with a CM10 transmission electron microscope (Philips, Eindhoven, The Netherlands).

## 5. Conclusions

In conclusion, our study has confirmed and demonstrated the presence of STCs in both MD and adjacent DT cells by immunohistochemical markers, and suggests that MD is a tissue undergoing non-negligible cell turnover. Since the origin of STCs still remains unknown, experiments aiming to clarify this issue are still needed. Understanding the origin of the STCs, the transcriptional factors, and the pathways that regulate them would have therapeutic implications in the treatment of tubular damage and the control of blood pressure.

## Figures and Tables

**Figure 1 ijms-23-10504-f001:**
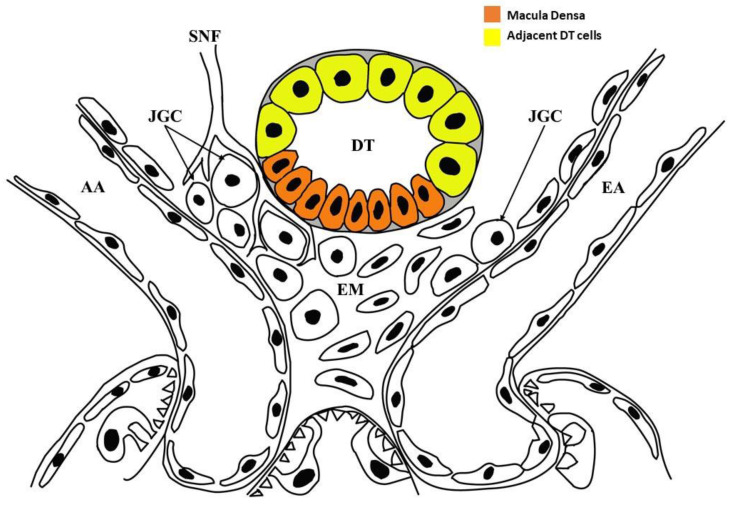
Schematic drawing of the juxtaglomerular apparatus. MD is depicted in orange and DT cells adjacent to MD in yellow. AA, afferent arteriole; DT, distal tubule; EA, efferent arteriole; EM, extraglomerular mesangium; JGC, juxtaglomerular cells; SNF, sympathetic nerve fibers. The figure has been created by Adobe Illustrator software (version 26.4.1; Adobe Inc., San Jose, CA, USA).

**Figure 2 ijms-23-10504-f002:**
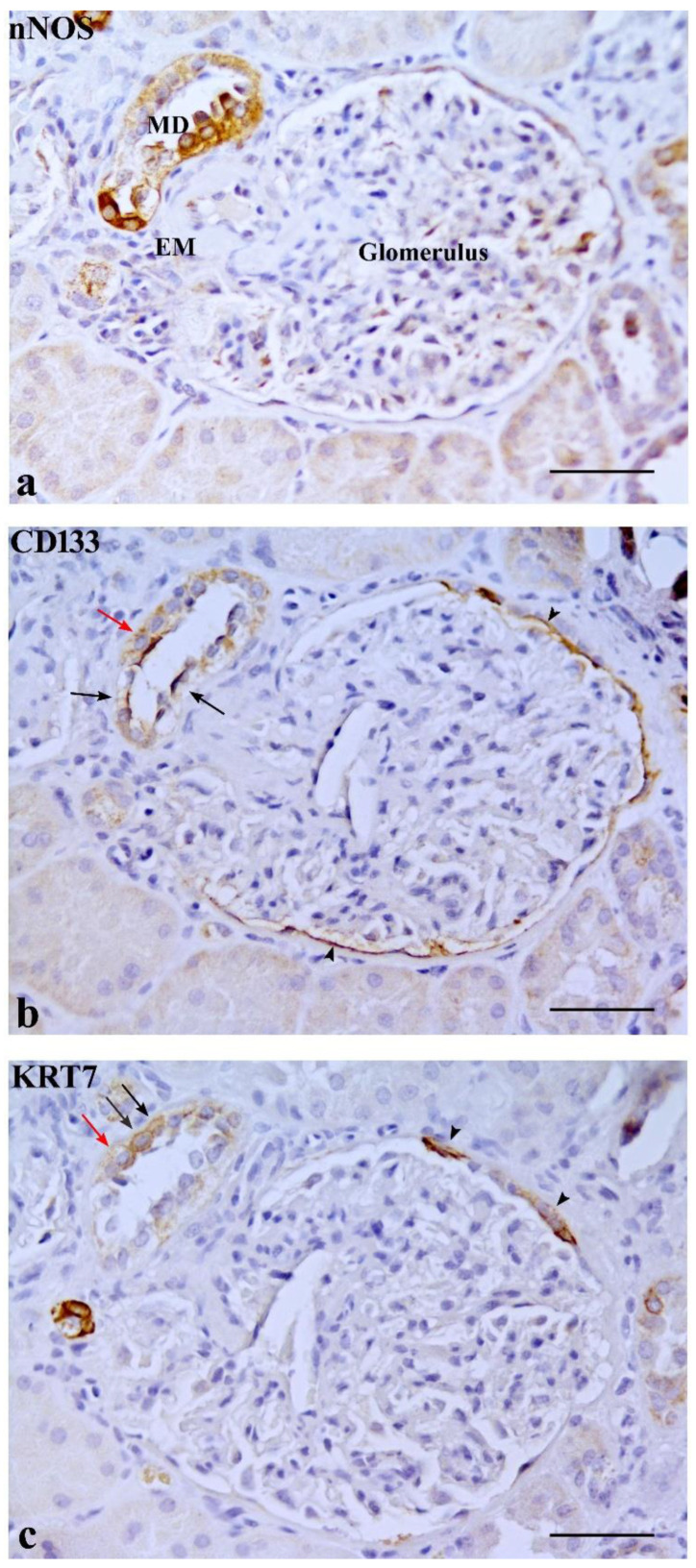
Three serial sections of the same macula densa (MD) stained with nNOS (**a**), CD133 (**b**) and cytokeratin 7 (KRT7) (**c**). (**b**) Some CD133-positive cells are localized in both MD and adjacent DT cells (arrows). The parietal epithelium of the Bowman’s capsule is also positive (positive internal control) (arrowheads). (**c**) KRT7 is localized in three cells (arrows) belonging to DT cells adjacent to MD as well as in two tracts of the parietal epithelium of the Bowman’s capsule (arrowheads). A cell of the DT co-expresses both CD133 and KRT7 markers (red arrows). EM, extraglomerular mesangium. Scale bar: 50 µm.

**Figure 3 ijms-23-10504-f003:**
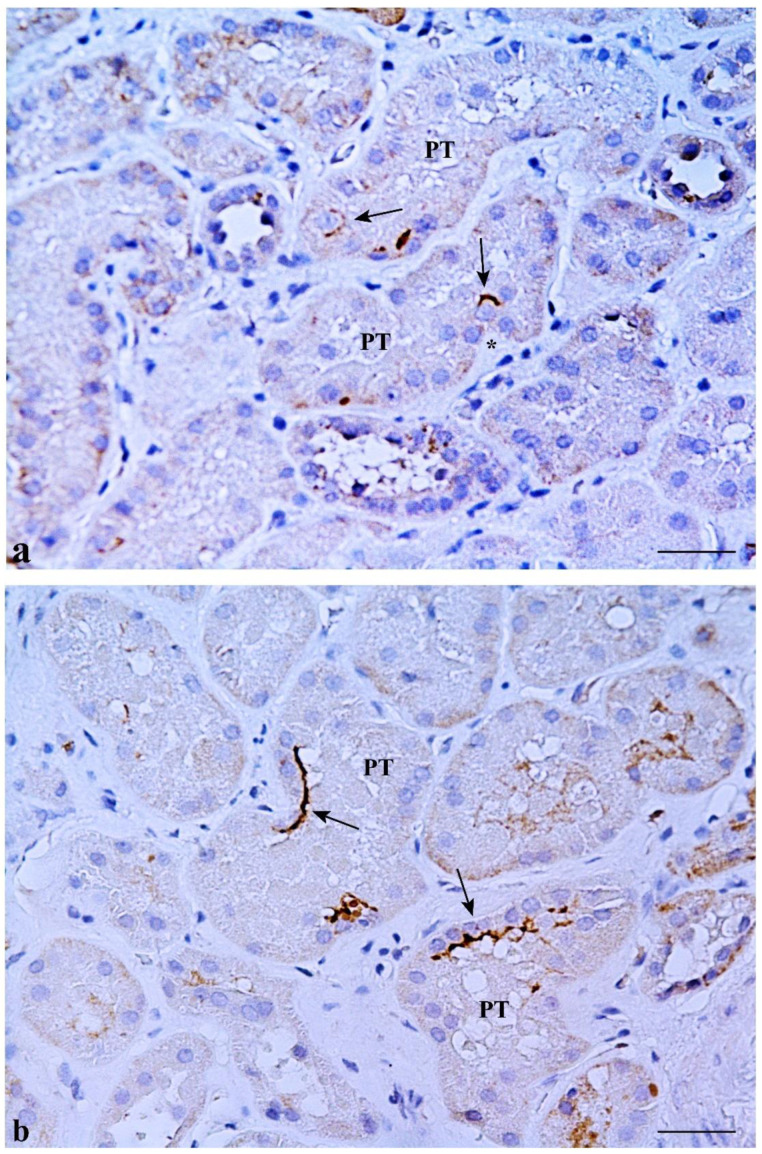
CD133 immunostaining in renal tissue. (**a**) Isolated cells expressing CD133 are present in two different proximal tubules (PT) (arrows). The apical staining appears in an arch shape. Note a CD133-positive cell showing an immune-stained arch in a tubular plicae (*). (**b**) CD133-positive cells are present in clusters in two PT (arrows). Scale bar: 100 µm.

**Figure 4 ijms-23-10504-f004:**
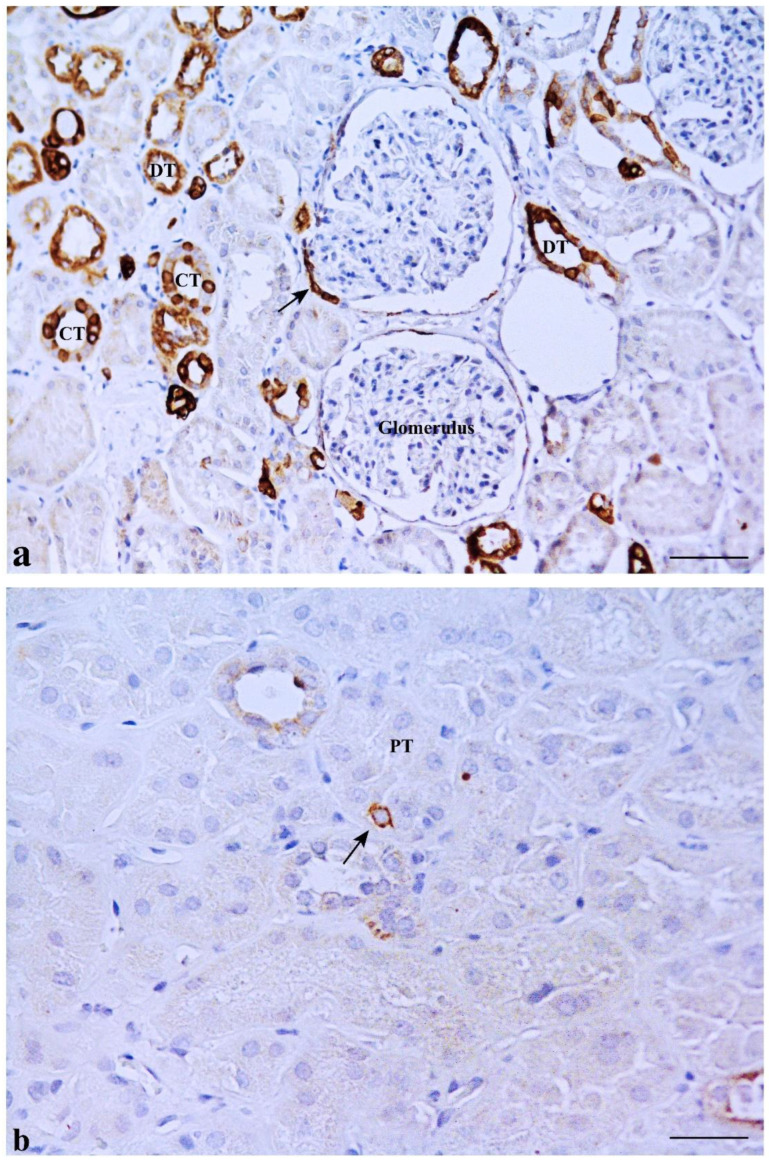
KRT7 immunostaining in renal tissue. (**a**) KRT7 positivity is mainly limited to distal tubules (DT) and collecting tubules (CT). Short segments of the parietal epithelium of Bowman’s capsule are rarely positive for KRT7 (arrow), while renal corpuscles are usually negative. (**b**) A single KRT7-positive cell is shown in a proximal tubule (PT) (arrow). Scale bar: 100 µm.

**Figure 5 ijms-23-10504-f005:**
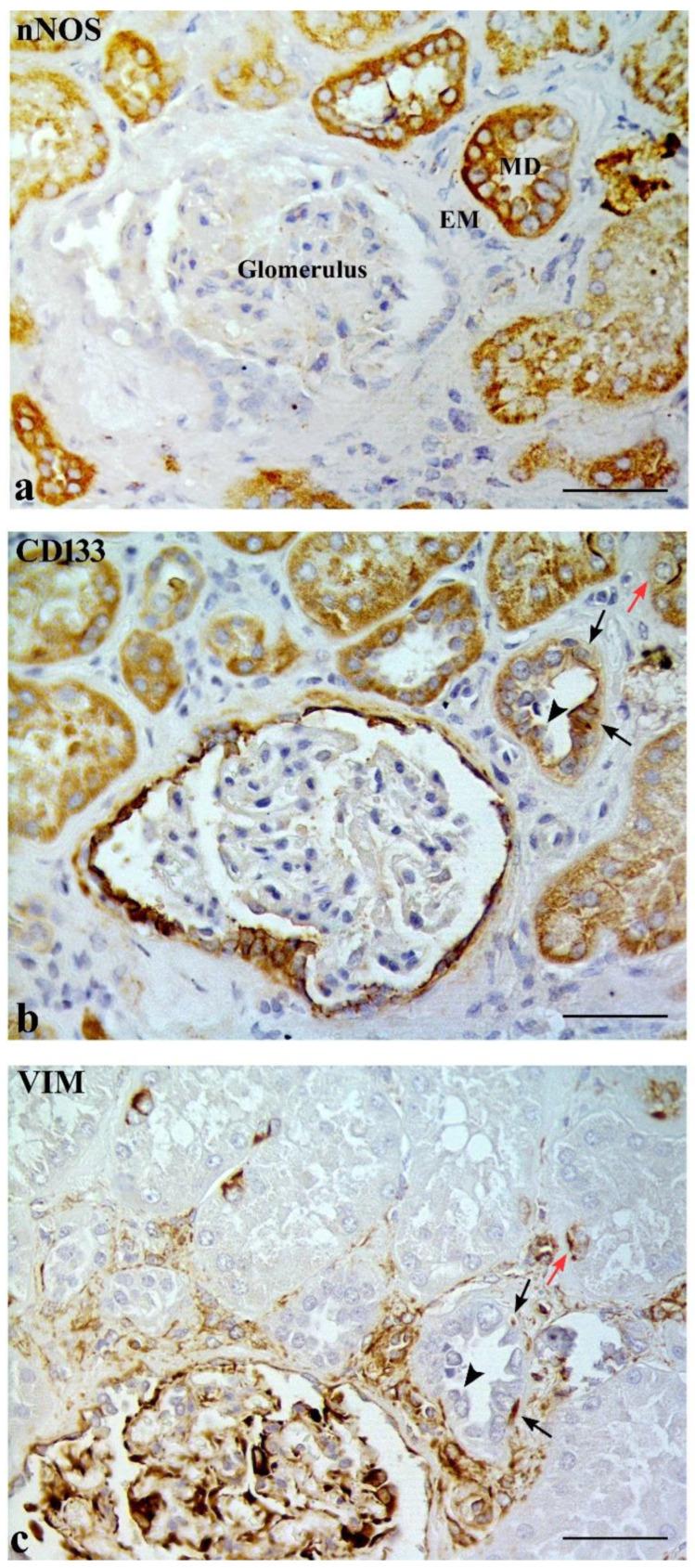
nNOS (**a**), CD133 (**b**) and VIM (**c**) expression in three serial sections of the same macula densa (MD). Two CD133-positive cells are localized in DT cells adjacent to MD (black arrows). The same cells are also positive for VIM in the basal pole (black arrows in (**c**)). The CD133-negative cell in MD (black arrowhead in (**b**)) is VIM-positive (black arrowhead in (**c**)). CD133 (red arrow in (**b**)) and VIM (red arrow in (**c**)) are co-expressed in a tubular cell. EM, extraglomerular mesangium. Scale bar: 50 µm.

**Figure 6 ijms-23-10504-f006:**
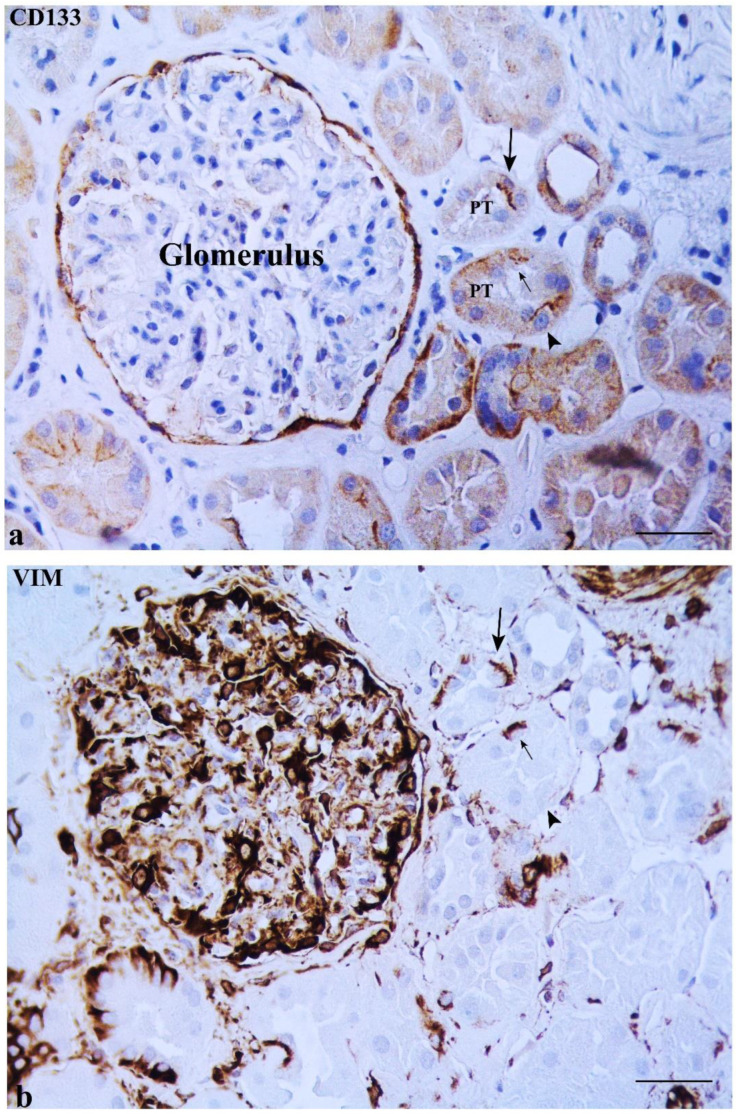
Expression of CD133 and VIM in renal tissue. (**a**) A few cells in a PT showed CD133 arch-shape immunostaining (big arrow in (**a**)) and VIM expression in the cellular basal pole (big arrow in (**b**)). A CD133-negative (small arrow in (**a**)) but VIM-positive cell (small arrow in (**b**)) as well as another CD133-positive (arrowhead in (**a**)) but VIM-negative (arrowhead in (**b**)) cell are present in another PT. Moreover, both the renal corpuscle (parietal epithelium and glomerulus) and interstitium are positive for VIM. Scale bar: 50 µm.

**Figure 7 ijms-23-10504-f007:**
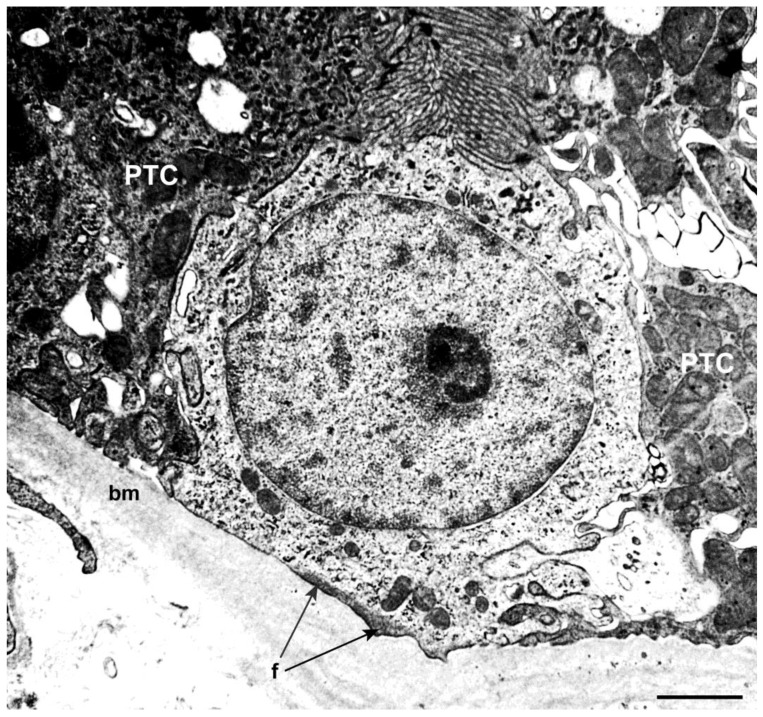
Ultrastructural image of an STC between two cells of the proximal tubule (PTC). Note the undifferentiated aspect and the presence of basal filaments (f: showed by the arrows). bm, basement membrane. Scale bar: 1 µm.

**Figure 8 ijms-23-10504-f008:**
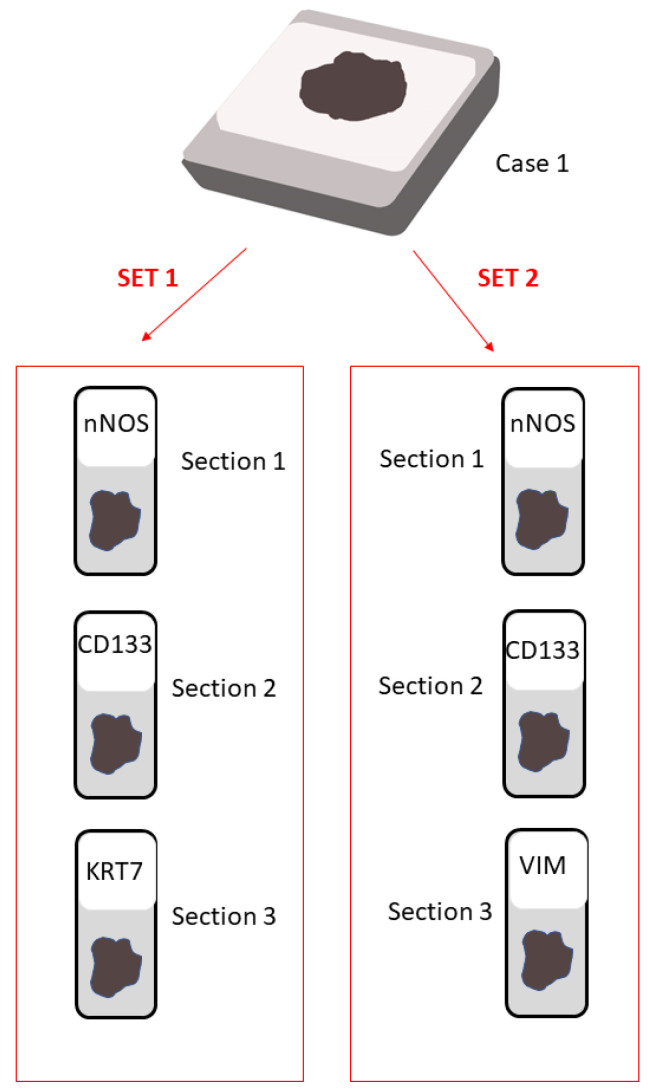
A schematic representation of the serial section setting. We prepared two sets of 3 serial sections (4 µm each one) in order to be sure to obtain at least a part of the macula densa (MD) (since it is arranged in 5–6 layers, extending for about 40 µm [14]). Moreover, in order to ensure the presence of MD in the section, the first sections of each set were stained for nNOS (a specific marker of MD) while the second sections of each set were stained for CD133. The third sections were stained for Cytokeratin 7 (KRT7, in set 1) and vimentin (VIM, in set 2). The figure has been created by Adobe Illustrator software (version 26.4.1; Adobe Inc., San Jose, CA, USA).

**Table 1 ijms-23-10504-t001:** Characteristics of tissue samples studied.

Cases	Gender	Age	Tumor Histopathology
Case 1	Female	78	Clear cell carcinoma
Case 2	Male	51	Clear cell carcinoma
Case 3	Male	85	Clear cell carcinoma
Case 4	Female	57	Oncocytoma
Case 5	Male	47	Clear cell carcinoma
Case 6	Female	37	Chromophobe renal cell carcinoma

**Table 2 ijms-23-10504-t002:** Antibodies used in this study.

Antibodies	Dilution	Company
Rabbit polyclonal nNOS (#160870)	1:500	Cayman Chem., Ann Arbor, MI, USA
Rabbit polyclonal CD133 (#HPA004922)	1:650	Sigma, St. Louis, MO, USA
Mouse monoclonal Cytokeratin 7 (#M7018)	1:50	Dako, Glostrup, DM, USA
Mouse monoclonal Vimentin (#GA630)	1:1	Dako, Glostrup, DM, USA
Isotype mouse IgG (#I-2000-1)	1:50	Vector lab., Burlingame, CA, USA
Isotype rabbit IgG (#I-1000-5)	1:500 or 1:650	Vector lab., Burlingame, CA, USA
Secondary goat anti-rabbit antibody (#BA-1000-1.5)	1:200	Vector lab., Burlingame, CA, USA
Secondary goat anti-mouse antibody (#BA-9200-1.5)	1:200	Vector lab., Burlingame, CA, USA

## Data Availability

Data available upon request from the corresponding author, due to the institutional policies.

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
