# Peer review of "Scattered Tubular Cells Markers in Macula Densa of Normal Human Adult Kidney"

_ijms, 2022, doi:10.3390/ijms231810504_

Round 1

Reviewer 1 Report

The authors examined the presence of scattered tubular cells (STCs) in the macula of normal human kidneys by immunohistochemical staining for specific markers of STCs. While this is an interesting study, it raises several minor concerns, including the following:

1.       Figure 2: I don't see any red arrows indicating cells that co-express both CD133 and CKT7 markers.

2.       "CD" in Figure 3a should be "CT".

3.       Figure 5. If the authors want to show the coexistence of markers, they should show a picture of the same area as in Figure 2. Why is it that only c in Figure 5 has a different depicted area?

4.       "mb" in Figure 7 should be "bm" as described in the figure legend.

5.       It may be more convenient to use immunofluorescence staining to show molecular co-localization. Is there any reason why the authors did not perform the IF staining method?

Author Response

  1. Figure 2: I don't see any red arrows indicating cells that co-express both CD133 and CKT7 markers.

We modified the image

  1. "CD" in Figure 3a should be "CT".

Maybe the reviewer means the arrow in figure Fig. 4 a. We modified the image according to the reviewer comment.

  1. Figure 5. If the authors want to show the coexistence of markers, they should show a picture of the same area as in Figure 2. Why is it that only c in Figure 5 has a different depicted area?

The Fig. 5, representing the second set of 3 serial sections, is similar to Fig. 2. However, the MD observed in c is the same of that showed in a and in b. The area has been shifted in order to better show the proximal tubule containing a vimentin positive cell in basal pole.

  1. "mb" in Figure 7 should be "bm" as described in the figure legend.

We modified the image

  1. It may be more convenient to use immunofluorescence staining to show molecular co-localization. Is there any reason why the authors did not perform the IF staining method?

We agree with the reviewer but, unfortunately, we cannot perform immunofluorescence staining in our lab. Thus, we opted for serial section analysis which is a type of analysis still used in morphology laboratories and allow us to study the same cell populations (https://www.ncbi.nlm.nih.gov/pmc/articles/PMC7484744/).

Reviewer 2 Report

In this study, Tossetta et al aim to confirm the presence of scattered tubular cells (STCs) in normal human macula densa using specific STC markers CD133, CKT7 and vimentin. 

The article needs some serious work in order for it to be accepted. Here are just a couple of suggestions for the authors:

1. It would be better if the authors conduct other experiments to confirm their findings.

2. The current findings are established in normal kidney conditions, I would like to ask the authors what is the usefulness of their findings in the kidney disease context? Perhaps the authors can add a section to discuss this.

3. Minor revisions:

- At lines 32, 100, and 101 KRT7 was mentioned without previous explanation of what it means or what its function is. This abbreviation was mentioned in figure 2 C as KR7 and in the legend of the same figure as CKT7. Acronyms should be standardized in the whole manuscript.

- More attention should be paid to the English language of this manuscript (e.g., at line 32 "letter" should be "latter" ).

Author Response

  1. It would be better if the authors conduct other experiments to confirm their findings.

We thank the reviewer for the suggestion. We plan to perform other experiments with more markers (e.g. CD24) to further validate our findings.

  1. The current findings are established in normal kidney conditions, I would like to ask the authors what is the usefulness of their findings in the kidney disease context? Perhaps the authors can add a section to discuss this.

We agree with the reviewer. We discussed possible applications of our findings in the discussion section. In particular, the results of our study showed for the first time the presence of stem cells in this very small tissue (macula densa) opening new prospective in the study of this tissue in renal diseases (lines

  1. Minor revisions:

- At lines 32, 100, and 101 KRT7 was mentioned without previous explanation of what it means or what its function is. This abbreviation was mentioned in figure 2 C as KR7 and in the legend of the same figure as CKT7. Acronyms should be standardized in the whole manuscript.

We standardized the acronyms in the whole manuscript as recommended by the reviewer

- More attention should be paid to the English language of this manuscript (e.g., at line 32 "letter" should be "latter" )

We checked the english in the whole manuscript as suggested

Round 2

Reviewer 2 Report

The manuscript has now improved significantly and the changes made are adequate.